# High β-Glucan Whole Grain Barley Reduces Postprandial Glycemic Response in Healthy Adults—Part One of a Randomized Controlled Trial

**DOI:** 10.3390/nu17030430

**Published:** 2025-01-24

**Authors:** Julianne A. Kellogg, Pablo Monsivais, Kevin M. Murphy, Martine M. Perrigue

**Affiliations:** 1Department of Crop and Soil Sciences, College of Agricultural, Human, and Natural Resource Sciences, Washington State University, Pullman, WA 99164, USA; julianne.kellogg@gmail.com (J.A.K.); kmurphy2@wsu.edu (K.M.M.); 2Department of Nutrition and Exercise Physiology, Elson S. Floyd College of Medicine, Washington State University, Spokane, WA 99202, USA; p.monsivais@wsu.edu

**Keywords:** whole grains, functional foods, biofortification, barley, beta-glucan, fiber, glycemia, satiety, appetite

## Abstract

Background/Objectives: The effects of sweetened and unsweetened high β-glucan whole grain barley on postprandial blood glucose response in normoglycemic human subjects were evaluated in a randomized, controlled, crossover clinical trial. Methods: Sixteen healthy, over-night fasted participants were studied on four or eight separate occasions. Participants consumed an unsweetened preload condition (*n* = 16): white glutinous rice (WR; 0 g β-glucan), low β-glucan barley (LB; ~4 g), medium β-glucan barley (MB; ~5 g), or high β-glucan barley (HB; ~6 g); or a sweetened condition with high fructose corn syrup (HFCS; *n* = 8): WR + 50 g HFCS, LB + 50 g HFCS, MB + 50 g HFCS, or HB + 50 g HFCS. After consuming the preload as a breakfast food, participants self-administered blood glucose tests every 15 min for four hours. Results: In both sweetened and unsweetened conditions, higher β-glucan content was associated with lower blood glucose peak response and incremental area under the curve estimates (iAUC). In comparison to the unsweetened conditions, the sweetened conditions resulted in less prominent decreases in mean blood glucose response and iAUC blood glucose as β-glucan content increased. Conclusions: By attenuating postprandial glycemic response, high β-glucan whole grain barley foods could play a role in helping to control blood glucose.

## 1. Introduction

In the United States (US), more than one in three adults has prediabetes and one in ten has diabetes, with 90–95% of diagnosed diabetes cases being type 2 diabetes [1]. Prediabetes is the condition of having higher than normal blood glucose levels, but lower than diabetes thresholds. Over 80% of the US population living with prediabetes is unaware of their condition [2]. High fasting blood glucose is one of the leading risk factors for years lived with disability or injury (YLD) in the US [3]. Prediabetes is associated with an increased risk of developing not only diabetes, but also kidney and nerve damage and cardiovascular disease [4,5,6].

Lifestyle modification is the foundation of diabetes prevention. Recommended dietary interventions, such as Mediterranean-style diets and the Dietary Approaches to Stop Hypertension (DASH) diet, include increased consumption of whole grains and are associated with a lower risk of developing type 2 diabetes [7]. Dietary fiber in whole grains may significantly reduce diabetes risk [8]. Such reductions in risk may be due in part to fiber characteristics, including viscosity and fermentability, that result in slowed glucose response [9].

In the US, dietary fiber is one of four dietary components of public health concern; dietary fiber is considered underconsumed by more than 90 percent of women and 97 percent of men [10]. Recommendations to consume 14 g of dietary fiber per 1000 kcal consumed are based on levels observed to reduce risk of coronary heart disease [10]. In the US, grain-based foods contribute 54.5% of all dietary fiber consumed [11]. Breeding crops to be higher in a nutrient (i.e., biofortification) can be a cost-effective and impactful nutrition intervention [12,13]. US and international barley breeding programs are increasing the fiber content of barley, specifically β-glucan content, as part of the efforts to increase fiber consumption [14,15,16,17]. β-glucan is a linear, unbranched, non-starchy polysaccharide composed of D-glucose monomers linked by (1, 3), (1, 4), or (1, 6) β-glycosidic bonds. Of all cereal crops that contain β-glucan, barley has the highest reported content [18]. Barley β-glucan, a mixed linked (1, 3), (1, 4)-β-d-glucan, becomes a viscous solution in the stomach and subsequently ferments in the colon [18]. Clinical trials have shown consumption of barley β-glucan to reduce total and low-density lipoprotein (LDL) cholesterol [19]; positively alter gut microbiota [20]; control appetite and reduce energy-intake [21]; and reduce postprandial blood glucose and insulin response [22].

To better understand the effect of barley β-glucan on blood glucose and insulin response, trials have investigated dose–response relationships [23,24,25]. These studies used barley derivatives (e.g., concentrates or flour fractions) or barley as an ingredient in a test meal. Varietal differences have not been investigated as the source of the dose in human clinical trials. The high degree of heterogeneity among clinical trials investigating barley β-glucan and postprandial glycemic response results in a lack of consensus among the findings [22]. Without such a consensus, it is not possible to issue a reviewed, marketable health claim for the potential of barley to modulate glycemic response. Currently, the only approved US Food and Drug Administration (FDA) health claim for food products containing β-glucan relates to the lowered risk of coronary heart disease [26].

Thus, well-designed clinical trials are needed to better understand the glycemic effect of high β-glucan whole grain barley foods. To determine the response of different high β-glucan barley varieties served as a breakfast food on postprandial blood glucose, participants consumed a porridge preload of an unsweetened condition: white glutinous rice (WR; 0% β-glucan), low β-glucan barley (LB; 7% β-glucan), medium β-glucan barley (MB; 8% β-glucan), or high β-glucan barley (HB; 10% β-glucan). The hulless food barleys, all with similar starch characteristics that were tested are as follows: ‘Havener’ (LB; Washington State University); ‘Meg’s Song’ (MB; Washington State University); and a commercially available variety (HB). To assess the ability of barley β-glucan to attenuate glycemic response to ingestion of high-sugar foods, participants consumed a sweetened condition with high fructose corn syrup (HFCS): WR + 50 g HFCS, LB + 50 g HFCS, MB + 50 g HFCS, or HB + 50 g HFCS.

## 2. Materials and Methods

### 2.1. Grain Characteristics

All grains included in this trial were milled by cereal chemists at the United States Department of Agriculture (USDA) Western Wheat Quality Lab (Pullman, WA, USA). A Perten mill (hammer-type cyclone mill) was used to mill grains to a maximum flour particle size of 0.8 mm. All grains in this trial were selected to be waxy (amylose-free) or near-waxy (low amylose content; <5%) [27]. Barley and rice were analyzed for gross energy (bomb calorimeter, model C5001, IKA, Staufen, Germany); protein (Nitrogen analyzer, model FP-528, LECO, St. Joseph, MI, USA); total dietary fiber (enzymatic and gravimetric assay, Sigma, St. Louis, MO, USA); and β-glucan (enzymatic assay kit, Megazyme, Bray, Ireland) (Table 1).

### 2.2. Experimental Preload Conditions

Preload conditions were standardized for volume (240 mL) and energy content (250 kcal unsweetened; 391 kcal sweetened) (Table 1). The preload conditions included unsweetened conditions (Unsweet) of white glutinous rice (WR; 0 g β-glucan), low β-glucan barley (LB; 4.2 g β-glucan), medium β-glucan barley (MB; 4.8 g β-glucan), and high β-glucan barley (HB; 5.7 g β-glucan); and sweetened conditions (Sweet) with high fructose corn syrup (HFCS; IsoClear^®^ 42%, Cargill, Minneapolis, MN, USA): WR + 50 g HFCS, LB + 50 g HFCS, MB + 50 g HFCS, and HB + 50 g HFCS. Preloads were prepared 10 min prior to serving by hydrating the pre-milled flour with boiling water and mixing vigorously to achieve a smooth porridge consistency. Preloads were served in a bowl with a spoon, alongside 355 mL of water in a cup to drink.

### 2.3. Trial Design

This study was conducted in accordance with the Declaration of Helsinki and the trial design, was approved by the Institutional Review Board of Washington State University (protocol code 17735) and was registered at clinicaltrials.gov as NCT06146322. This trial utilized a randomized, controlled, two-factor crossover design. Eight experimental preload conditions were administered to each participant over eight weeks of testing with minimum five-day washout periods. On visits 1, 2, 3, and 4, participants consumed one of four unsweetened preloads: WR, LB, MB, and HB. On visits 5, 6, 7, and 8, participants consumed one of four preloads sweetened with HFCS: WR + 50 g HFCS, LB + 50 g HFCS, MB + 50 g HFCS, and HB + 50 g HFCS.

At each visit, participants arrived after an overnight fast and completed a check-in survey on an electronic portable tablet (Galaxy Tab A, Samsung, Suwan-si, Republic of Korea) to verify continued eligibility. Using the check-in survey, participants verified that they had been fasting since 20:00 h the night before, that they had not used any nicotine or marijuana products, or illicit drugs since their last appointment, enrolled in any other clinical studies since their last appointment, or been diagnosed with any new medical condition or started any new medication since their last appointment. In order to eliminate potential differences in energy needs and daily glycemic profiles across study conditions, participants were required to maintain a consistent level of physical activity throughout their study enrollment. Using the check-in survey, participants verified on each study day that they had not engaged in any physical activity outside of their normal routine. At 08:00 h, participants self-reported their baseline (fasting) appetite ratings using computerized 100 mm Visual Analog Scales (VASs) on an electronic tablet and tested their baseline blood glucose using a finger prick and a FreeStyle Freedom lite glucometer (Abbott, Chicago, IL, USA). Next, the preloads and 355 mL of water were served on a tray. Participants were allowed 15 min to consume all food and drink. Researchers collected trays and verified complete consumption of preload and water at 08:15 h. Every 15 min thereafter, participants self-reported appetite ratings and tested their blood glucose. A total of 17 finger pricks (one every 15 min) were collected from each participant between 08:00 h and 12:00 h. After the final blood glucose measurement and appetite rating at 12:00 h, participants were given a test lunch. Participants were instructed to eat ad libitum. At 12:30 h, participants completed a final appetite rating.

### 2.4. Measurements

To measure blood glucose, participants used a disposable safety lancet to collect a capillary blood drop from their fingers and a point-of-care glucometer. Participants used the same glucometer at each visit, unless a malfunction in that glucometer was discovered. A manufacturer-provided control solution was used prior to each testing session to ensure proper calibration of each glucometer. The use of a point-of-care glucometer is uncommon in a human clinical trial to assess glycemic response. In other studies, blood draws are typically more intrusive but occur less frequently, with blood being drawn with an in-dwelling venous catheter only five to nine times over the course of two to three hours [23,24,25,28,29,30]. In the present study, seventeen non-invasive finger pricks were completed by each participant over the course of 4 h; the greater number of collections allowed for a better chance to observe peak blood glucose.

### 2.5. Participant Sample Size Determination

A sample size for a continuous outcome with equal sample sizes in each group was determined using 2 h postprandial blood glucose incremental area under the concentration time curve (iAUC) data from a study investigating low, medium, and high β-glucan oat food products (2 g, 4 g, and 6 g β-glucan) [29]. The present study was powered to detect differences in the primary outcome, blood glucose response. Alpha (α) = 0.05 and power = 0.80 (β = 0.20) were chosen as desirable error rates. The required sample size was determined to be 23 participants and the study aimed for a minimum of 32 enrolled participants. Our study fell short of the desired sample size because the trial had to be terminated early due to COVID-19 restrictions, as explained in Section 2.8 below.

### 2.6. Participants

Eligibility criteria were intended to select a set of participants homogeneous with respect to age and health characteristics to minimize potential confounding of results (Table A1). Eligible participants were healthy men and women 18–50 years of age with Body Mass Index (BMI) values of 18.5–40.0 kg/m^2^ with normal fasting blood glucose (<100 mg/dL). Participants were given an orientation and a training on monitoring their blood glucose with a glucometer. After giving informed consent for the trial, participants were randomly assigned to one of four sequences of preload conditions constructed using a Williams Design. Eligible participant enrollment was dependent on vacancy in the trial, resulting in continuous enrollment. Participants were given the option to participate in only four weeks of the trial; if they chose the four-week trial, they consumed only the unsweetened preloads.

### 2.7. Data Management, Calculations, and Statistical Analyses

Data were collected and managed with Research Electronic Data Capture (REDCap) tools hosted by Washington State University [31,32]. iAUC for the primary outcome, blood glucose, was calculated over multiple periods (1–4 h) according to the trapezoidal rule; the area below the baseline fasting levels was ignored [33,34]. Single imputation using the mean of prior and post measurements was computed for one missing data point that interfered with computing iAUC for blood glucose for one participant.

Statistical analyses were conducted with R (version 4.0.3) and the lme4 and emmeans packages [35,36,37]. Data were fitted to linear mixed effects models using restricted maximum likelihood with preload (four levels), sweetness (two levels), period (eight levels), and gender (two levels) as fixed effects and participants as a random effect. Data met assumptions for linearity, homogeneity of variance, or normality. Estimated marginal means (also known as least-squares means) were computed to conduct pairwise comparisons among all porridge preloads for the primary and secondary outcomes. A difference was regarded as significant if the *p*-value was <0.05. Confidence intervals (95% CI) were computed; data were compatible with any value of μ within the CIs but relatively incompatible with any value outside the CIs [38].

### 2.8. Changes in Response to the COVID-19 Pandemic

For studies affected by the pandemic, researchers are encouraged to describe changes to the study protocol, trial interruptions, missing data, and statistical power [39]. All research activities at Washington State University were suspended mid-March 2020 to control the spread of COVID-19. The protocol remained unchanged, but the trial was terminated early. Thus, the study was unable to reach the desired sample size. The early termination of the trial resulted in an unbalanced experimental design with more participants consuming unsweetened porridges than sweetened porridges (Table 2). Statistical methods account for the unbalanced design. A linear mixed effects model was used because it is an appropriate statistical tool for cross-over trials with incomplete data [40,41]. The original research hypotheses were tested with the initially intended pairwise comparisons across all porridge preloads.

Early termination of the trial resulted in missing data. Missing data were considered missing completely at random (MCAR) for all but one participant in the resulting data set. The MCAR designation was justified for the participants with known reasons for early termination of the trial (*n* = 10). Such reasons included the following: (1) the participant signed up for only the first four weeks of the trial; (2) the participant notified the researchers of impending ineligibility; and (3) the participant was only partly through the trial when COVID-19 research restrictions caused the trial to conclude early. One participant with missing data was excluded from the analysis because researchers were not notified of the reason for trial termination.

## 3. Results

### 3.1. Sample

Of the 30 persons allocated to an intervention (Figure 1), 16 healthy adults were included in the final analysis (Table 3).

### 3.2. Blood Glucose

Differences in blood glucose response among porridge preloads were greatest at the early timepoints of 15, 30, 45, and 60 min, with diminishing differences at later timepoints (Figure 2). Peak blood glucose response was at 30 min; the white rice (WR) control resulted in the highest mean peak blood glucose response in both the sweetened (Sweet) and unsweetened (Unsweet) preload conditions. In both the sweetened and unsweetened conditions, β-glucan content was inversely related to mean peak blood glucose.

In the unsweetened conditions, the low β-glucan (LB), medium β-glucan (MB), and high β-glucan (HB) barley preloads resulted in lower mean peak blood glucose than WR. The confidence intervals at peak blood glucose for unsweetened LB and MB did not overlap with WR but overlapped with each other. The unsweetened HB preload confidence intervals did not overlap with any other unsweetened preload at peak blood glucose. In the unsweetened conditions, WR resulted in a significant postprandial reduction in blood glucose (*p* < 0.05); blood glucose response was below baseline at 180 min and confidence intervals did not overlap with the barley preloads.

In the sweetened conditions, MB and HB peak blood glucose was significantly lower than WR peak blood glucose (confidence intervals did not overlap). However, there were no differences among the barley preloads. At baseline (0 min), there was higher variability in the blood glucose data in the sweetened conditions than in the unsweetened conditions; the higher variability is likely a result of the low number of participants that consumed the sweetened conditions (Figure 2).

The most evident differences in iAUC blood glucose response data were found among porridge preloads for the 1 h iAUC period (Figure 3a). In the unsweetened conditions, all barley preloads resulted in significantly lower 1 h iAUC blood glucose than the WR control (*p* = <0.001). In the unsweetened conditions, the LB and MB barley preloads were not significantly different from each other (*p* = 0.162), but HB resulted in significantly lower 1 h iAUC blood glucose than both LB and MB (*p* = <0.001). In the sweetened conditions, MB and HB resulted in significantly lower 1 h iAUC blood glucose than WR (*p* = <0.001), but LB did not differ from WR (*p* = 0.069). Consumption of sweetened LB and HB resulted in significantly lower iAUC blood glucose than their sweetened counterparts (*p* = 0.034 and *p* < 0.001, respectively).

For the subsequent hour estimates of 2, 3, and 4 h iAUC blood glucose, the barley preloads resulted in significantly lower iAUC than WR in the unsweetened conditions (Figure 3b–d). In the sweetened conditions, LB was not significantly different from WR at 2, 3, and 4 h iAUC; MB and HB resulted in significantly lower 2 h and 3 h iAUC blood glucose than WR but were not statistically significantly different from WR at 4 h iAUC blood glucose.

Differences in blood glucose response among the barley preloads were most evident where the differences in β-glucan content were greatest. The iAUC blood glucose at 2, 3, and 4 h for unsweetened HB were significantly lower than the iAUC estimates for unsweetened LB and MB; sweetened, HB was not significantly lower than LB or MB. There were no significant differences in iAUC blood glucose between LB and MB in the sweetened or unsweetened conditions at 2, 3, and 4 h iAUC. Comparing the sweetened and unsweetened conditions within a preload, the greatest differences were between the sweetened and unsweetened conditions of HB (2 h: *p* = 0.02; 3 h: *p* = 0.052; and 4 h: *p* = 0.069).

## 4. Discussion

The study was initially powered to detect differences in blood glucose response among the preloads in the unsweetened conditions. Although the ideal sample size of 24 participants was not achieved, the magnitude of the treatment differences was great enough to result in statistically significant differences among the preloads in the unsweetened conditions [42]. In the unsweetened conditions, differences in blood glucose response among the preloads were most evident where differences in β-glucan content were greatest. Fewer differences in blood glucose were observed in the sweetened conditions, conditions that had fewer data points. The lack of statistically significant differences could be due to small effect sizes or high variability in the data [43].

The effect of unsweetened whole grain high β-glucan barley foods on postprandial blood glucose is promising; in the present study, peak mean blood glucose response and iAUC for multiple periods was inversely related to barley β-glucan content. The barley foods made from the barley varieties ‘Havener’ (~4 g β-glucan), ‘Meg’s Song’ (~5 g β-glucan), and the commercially available variety (~6 g β-glucan) resulted in lower 2 h iAUC blood glucose than the glutinous white rice control (0 g β-glucan) in the unsweetened conditions. It is not surprising that the barley preload lowest in β-glucan (~4 g) elicited a decrease in glycemic response; a 2013 review of human studies that investigated postprandial blood glucose response to whole grain and processed oat and barley foods found 4 g of β-glucan can significantly reduce postprandial blood glucose [44].

Comparatively, a study with a similar design, but with 17 women with increased risk for insulin resistance, tested hot breakfast cereals formulated using wheat and the barley variety Sustagrain^®^ (0, 2.5, 5, 7.5, 10 g β-glucan) and found no differences among the preloads for 2 h AUC blood glucose [23]. Participant characteristics may account for the divergent results between the present study and the study by Kim et al. [23]. In a 2016 meta-analysis of randomized controlled studies that investigated the effect of barley β-glucan on postprandial glycemic response in healthy humans, results showed that consumption of barley and barley β-glucan lower postprandial blood glucose response [22]. However, because the meta-analysis included a low number of studies (*n* = 17) and there was substantial heterogeneity among the studies, the authors concluded that more high quality dose–response studies are required to verify that barley can improve glycemic outcomes in healthy humans [22].

For the present study, the sweetened conditions of the preloads resulted in less prominent decreases in mean blood glucose response and iAUC blood glucose for the multiple periods as β-glucan content increased. It appears that the present study is the first to compare barley food preloads in sweetened and unsweetened conditions. With the findings that high fructose corn syrup may reduce the beneficial impact of β-glucan on postprandial blood glucose response and considering many breakfast foods are high in sugar, more studies are required to understand the effect of barley β-glucan on postprandial blood glucose response in the context of food with higher sugar content.

In the present study, Body Mass Index was used as an eligibility criterion. It is known that BMI is not an accurate predictor of body fat percentage, which may impact glycemic control. A better estimate of body fat percentage, would have been determined using a Bioelectrical Impedence Analysis (BIA) or Dual X-ray Absorptiometry (DEXA) scan.

### To Whom Do These Results Apply?

Although the present study excluded individuals with prediabetes, type 1 diabetes, and type 2 diabetes, the results suggest whole grain barley foods high in β-glucan modulate glycemic response and can be a healthy food choice for normoglycemic individuals. The study was designed to achieve external validity while maintaining high internal validity. Inclusion criteria allowed for some heterogeneity in participant characteristics. Study participants had a wide range in BMI (18.6–31.7 kg/m^2^), with some that could be classified as ‘overweight’ or obese, and participant ages ranged from 20 47 years. Some of the study participants are part of the target population for prediabetes screening; the US Preventive Services Task Force (USPSTF) recommends screening for prediabetes and type 2 diabetes in adults aged 35 to 70 years who have overweight or obesity [45].

Point-of-care glucometers were used in the present study and such glucometers have been documented as having poor validity and reliability [46]. Although the glucometers may have negatively affected the internal validity of the study, their use improves the external or ecological validity of the results. The use of blood glucose measurement instruments that are used by the general population to track glycemia resulted in a data set approaching ‘real-world’ results. For further generalizability of study results, it is important to consider that the study occurred on a university campus with most participant recruitment occurring on campus and subgroup analysis was not conducted due to the low sample size.

Countries have robust regulations regarding nutrition and health claims. For the development of such claims, there must be a consensus of research evidence to support the claim. With the current lack of consensus on the potential of barley β-glucan to modulate glycemic response, it is not possible for countries to issue a reviewed, marketable health claim. Currently, the only approved U.S. Food and Drug Administration (FDA) health claim for food products containing β-glucan relates to the lowered risk of coronary heart disease. The findings from this study contribute to the body of literature on barley β-glucan and glycemic response and will facilitate the development of additional marketable barley health claims.

## 5. Conclusions

Unsweetened whole grain barley preloads (containing ~4, ~5, and ~6 g β-glucan) resulted in lower postprandial peak blood glucose compared with the white rice control (0 g β-glucan). Sweetened, the barley preloads highest in β-glucan (~5 and ~6 g β-glucan) resulted in lower postprandial peak blood glucose than the white rice control. The higher the β-glucan content of the preloads, the lower the peak blood glucose response. For 1, 2, 3, and 4 h iAUC blood glucose, the barley preloads resulted in significantly lower blood glucose iAUC than the control in the unsweetened conditions. For the sweetened conditions, the highest β-glucan barley preloads generally resulted in lower iAUC blood glucose than the control but differences among preloads were not as distinct, fewer participants consumed the preloads (*n* = ~8), and there was high variability in the data. Findings are in agreement with the current literature on the effect of barley β-glucan on blood glucose response and contribute to the evidence required for a postprandial glycemic reduction health claim.

## Figures and Tables

**Figure 1 nutrients-17-00430-f001:**
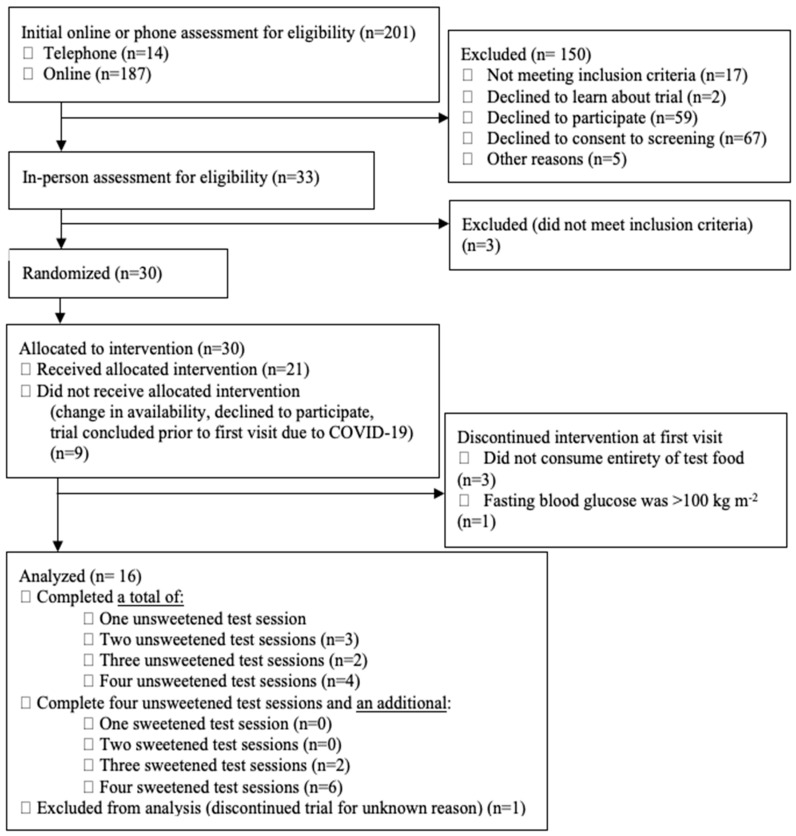
Consolidated Standards of Reporting Trials (CONSORT) flow diagram displaying participant allocation to trial treatments and participant retention.

**Figure 2 nutrients-17-00430-f002:**
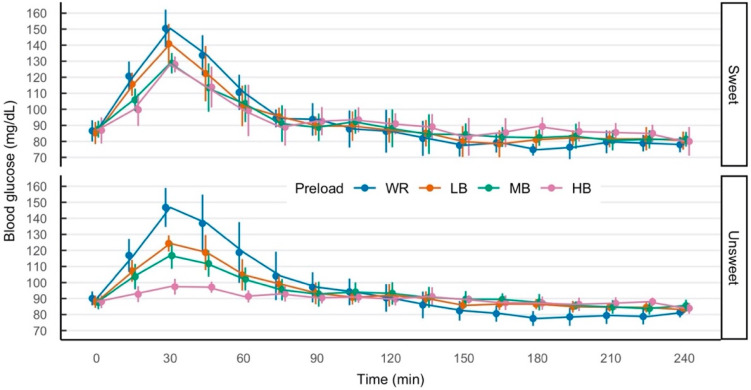
Temporal profiles of point means and confidence intervals (95%) for participant blood glucose responses as a function of preload condition. Porridge was served immediately after baseline assessment (0 min) and the test lunch was served after the 240 min assessment. WR, white glutinous rice, 0 g β-glucan; LB, low β-glucan barley, ~4 g β-glucan; MB, medium β-glucan barley, ~5 g β-glucan; HB, high β-glucan barley, ~6 g β-glucan.

**Figure 3 nutrients-17-00430-f003:**
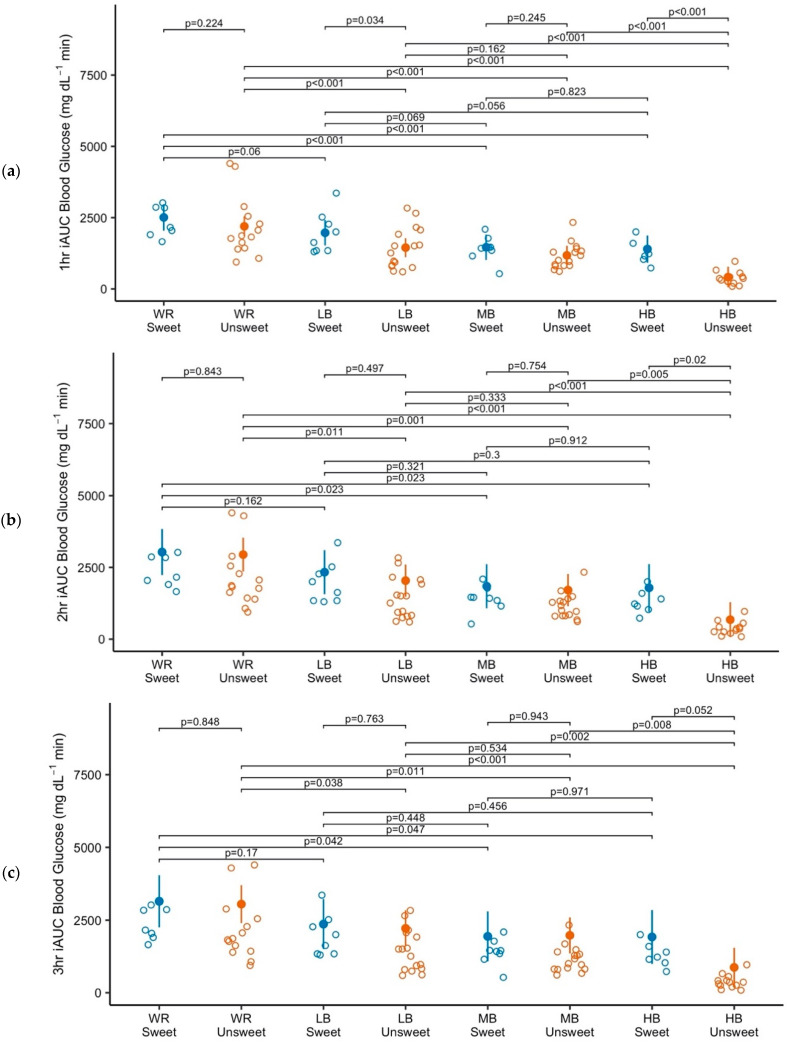
Data points, estimated marginal means, confidence intervals (95%), and pairwise comparison *p*-values for participant iAUC blood glucose as a function of preload condition at (**a**) 1 h; (**b**) 2 h; (**c**) 3 h; and (**d**) 4 h. Each *p*-value bracket represents a single comparison. WR, white glutinous rice, 0 g β-glucan; LB, low β-glucan barley, ~4 g β-glucan; MB, medium β-glucan barley, ~5 g β-glucan; HB, high β-glucan barley, ~6 g β-glucan.

**Table 1 nutrients-17-00430-t001:** Porridge content and nutrition facts for the white rice control (WR), low β-glucan barley (LB), medium β-glucan barley (MB), and high β-glucan barley (HB) in the sweetened (Sweet) and unsweetened (Unsweet) conditions.

Preload		β-Glucan (%, Dry Weight Basis)	β-Glucan (g)	Total Dietary Fiber (g)	Protein (g)	Flour Amount (g)	Energy (Kcal)	Volume (mL)
Sweet	WR	0	0.0	2.6	3.3	65.5	391	240
	LB	8.2	4.2	10.2	3.6	60.2	391	240
	MB	8.8	4.8	11.4	4.2	60.4	391	240
	HB	11	5.7	25.1	6.8	57.1	391	240
Unsweet	WR	0	0.0	2.6	3.3	65.5	250	240
	LB	8.2	4.2	10.2	3.6	60.2	250	240
	MB	8.8	4.8	11.4	4.2	60.4	250	240
	HB	11	5.7	25.1	6.8	57.1	250	240

**Table 2 nutrients-17-00430-t002:** Number of participants who consumed each test porridge, either sweetened with 50 g HFCS (Sweet) or unsweetened (Unsweet) ^1^.

	Preload Condition			
	WR	LB	MB	HB
Sweet	7	8	8	7
Unsweet	14	16	16	13

^1^ WR, white glutinous rice, 0 g β-glucan; LB, low β-glucan barley, ~4 g β-glucan; MB, medium β-glucan barley, ~5 g β-glucan; HB, high β-glucan barley, ~6 g β-glucan.

**Table 3 nutrients-17-00430-t003:** Participant characteristics by gender (n = 16). Data displayed as means and standard deviation (SD).

	Males (*n* = 7)	Females (*n* = 9)	Total (*n* = 16)
Age (y)	32.3 (9.1)	28.8 (8.1)	30.3 (8.4)
Body Mass Index (kg/m^2^)	24.4 (1.9)	24.6 (4.6)	24.4 (3.6)
Weight (kg)	81.0 (5.5)	66.7 (15.2)	72.9 (13.8)
Blood glucose at screening (mg/dL)	89.6 (6.5)	89.6 (7.5)	89.6 (6.8)

## Data Availability

The data presented in this study are available on request from the corresponding author. The data are not publicly available to protect participant confidentiality.

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
