# Peer review of "High β-Glucan Whole Grain Barley Reduces Postprandial Glycemic Response in Healthy Adults—Part One of a Randomized Controlled Trial"

_nutrients, 2025, doi:10.3390/nu17030430_

Round 1

Reviewer 1 Report

Comments and Suggestions for Authors

Authors present a trial on high glucan action on glycemic response, but there are several point that should be improved:

- Nothing is said about physical activity of participants, it is well known that it is a very important parameters in glycemic control even in diabetes condition see for example 10.14814/phy2.15740

- The only BMI is a poor indicator to define subjects, at least kinantropometry or better BIA should be performed

- It is unlceare which is the final goal of the project and the possible practical application, even I agree on lack of fibers on actual nutritional habits

Author Response

To whom it may concern:

Thank you for the reviewer comments on our manuscript.  We have addressed each comment here, and if needed, with changes to the manuscript. 

Kindly,

Martine Perrigue

Reviewer 2 Report

Comments and Suggestions for Authors

The manuscript presents a well-structured randomized controlled trial investigating the effects of high β-glucan whole grain barley on postprandial glycemic response in healthy adults. The study addresses an important topic in nutrition and diabetes prevention, and the findings contribute valuable insights into the role of dietary fiber in glycemic control.

My concern for this study is what is the basis of sample size selection for population studies? Whether the sample size of the population selected in this study is a little insufficient?

Why did the authors focus only on changes in blood sugar? Does the intake of intervention substances affect other health indicators? Such as security metrics?

Author Response

(The authors gave the same response as above.)

Reviewer 3 Report

Comments and Suggestions for Authors

Thank you for submitting the manuscript "High Β-Glucan Whole Grain Barley Reduces Postprandial Glycemic Response in Healthy Adults – Part One of a Randomized Controlled Trial" to Nutrients. 

The authors reported changes in the glycemic profile of individuals without diabetes after ingestion of oats with different levels of β-glucan. The manuscript is well written, the subject is interesting, but I have some considerations. 

Line #17: Is it not clear whether the authors did a "control treatment" (positive)?? But no control, only with the high fructose meal? 

Why was oats with normal β-glucan levels not used as a positive control? Rice may contain resistant starch that may not be identified by the dietary fiber content) unlike what happens with β-glucan. 

The introduction serves as a justification for the study. The researchers justify the work by the number of people with pre-diabetes in the US, but in other parts of the text (in the abstract, for example) it is stated that the individuals used in the study were normoglycemic. This needs to be reconsidered, especially in the introduction, since this is where the hypothesis of the work is presented. 

The introduction also needs to define the recognized parameters for pre-diabetes and diabetes in terms of definition. 

It is important to discuss the presence of other components in oats. Phenolic compounds, for example, were not even mentioned and are present in this raw material. If they are the same in quantity (previously reported in the Literature) in all samples, this also needs to be reported. 

The authors focus on discussing the amount of b-glucan, but there are other components in oats that make up dietary fiber that also influence the result and need to be discussed.

Author Response

(The authors gave the same response as above.)

Round 2

Reviewer 1 Report

Comments and Suggestions for Authors

The authors tried to answer my comments anyway, but some concerns remain.

- Physical activity is a central point, so it should be stated as a weakness

- The unique use of BMI should be highlighted as a weakness of the manuscript. I understand that DXA is a considerable cost, but BIA and/or kinanthropometry is not

- The purpose of the manuscript seems to be a marketing rather than a scientific purpose

Author Response

(The authors gave the same response as above.)
